# The Learnability of the Multiplayer Adversarial Bandit Problem

## Abstract

We study the multiplayer adversarial bandit problem under information asymmetry: $M$ players each select actions from their own action sets, forming a joint action, and observe (possibly noisy) losses. The players may agree on a strategy before learning begins but cannot communicate during play. We first show that when players act simultaneously, any strategy—including stochastic ones—suffers $\Omega(T)$ regret against an adaptive adversary, regardless of the number of players $M$ or arms $K$. To circumvent this impossibility, we introduce a sequential model in which players act in a fixed order, each observing the preceding players' choices. By establishing a connection between the resulting multiplayer policy and the classical single-player Hedge algorithm of Freund & Schapire (1997), we develop `mHedge-A`, which achieves $O(\sqrt{TM \log K})$ regret when all players observe the same losses, and `mHedge-B`, which achieves $\tilde{O}(M^2 \sqrt{T})$ regret under reward asymmetry with sub-Gaussian noise.

## 1 Introduction

The adversarial bandit problem has emerged as a foundational framework for sequential decision-making under uncertainty when faced with a potentially malicious environment. In this setting, a learner repeatedly selects one of $K$ actions (arms) and suffers a loss chosen by an adversary; the learner's goal is to minimize *regret*, defined as the difference between the learner's cumulative loss and that of the best fixed arm in hindsight. Unlike stochastic bandits, where rewards are drawn from fixed but unknown distributions, adversarial bandits consider scenarios where rewards may be chosen by an adaptive adversary who actively attempts to minimize the decision-maker's cumulative reward. This model has proven invaluable in applications ranging from online advertising and recommendation systems to network routing and security, where unpredictable or strategic environments naturally arise. The problem has been extensively studied since the seminal work of Auer et al. (1995), which established the EXP3 and Hedge algorithms achieving near-optimal regret bounds of $O(\sqrt{KT \log K})$ for $K$ arms over $T$ rounds. Subsequent research has further refined these bounds (examples include Audibert & Bubeck (2010) and Orabona & Pál (2016)), as well as developed algorithms with improved performance guarantees under various constraints and assumptions about the adversary's capabilities and the feedback mechanisms available to the learner (Bubeck & Slivkins (2012)).

In a different vein, many real-world applications inherently involve multiple decision-makers operating in shared environments. Numerous recent works have extended bandit frameworks in general to multiplayer settings Chang et al. (2022); Chang & Lu (2024); Chang & Karthik (2025); Chang & Lu (2025), where each player selects actions from their own action set, collectively generating joint actions and corresponding rewards. While the objective of multiplayer bandits remains similar to the single-player case, these settings introduce a fundamental challenge: *information asymmetry*. Depending on the problem formulation, players may observe joint actions but not other players' rewards (action asymmetry), receive different noisy observations of the same underlying events (reward asymmetry), or face both action and reward asymmetries. Addressing these various forms of information asymmetry represents the central challenge in multiplayer bandits research.

In this work, we study the adversarial bandit problem in a multiplayer setting with information asymmetry. We consider a version of multiplayer adversarial online learning where each agent has a distinct set of individual actions (marginal arms), and the resulting joint outcome is determined by the combination of

arms selected by all players. This formulation naturally models distributed decision-making scenarios where agents have different capabilities or action spaces. For example, in multi-robot coordination (Landgren et al., 2016), each robot may choose from distinct motion primitives, yet the team's overall performance depends on their joint configuration. Similarly, in decentralized resource allocation (Maghsudi & Stańczak, 2014; Anandkumar et al., 2011), distributed agents must partition shared resources without centralized coordination. Our approach extends the classical adversarial framework established by Auer et al. (1995) to the multiplayer domain, while building upon the current literature of multiplayer bandits as established by Chang et al. (2022). This setting also covers information asymmetry challenges, as players must coordinate despite having limited visibility into others' action spaces and observations. We defer additional related works to appendix A.

**Our contribution** In Chang et al. (2022); Chang & Lu (2024; 2025); Chang & Karthik (2025), the authors were able to prove near optimal regret bounds for the multiplayer information asymmetry stochastic bandit problem. Remarkably, we show that in the same setting, when the losses are adversarial, a strategic adversary can force the *players* to incur linear regret—that is, the players' cumulative loss exceeds that of the best joint action in hindsight by $\Omega(T)$. To address this, we modify the multiplayer setting so that actions occur sequentially, allowing each player to observe the choices of the previous one. In the case where all players experience the same losses, we propose `mHedge-A`, an algorithm that achieves a regret bound of $\tilde{O}(\sqrt{TM})$, where $T$ represents the number of rounds and $M$ the number of players. The key insight is that sequential conditioning, combined with Bayes' rule, allows the multiplayer policy to replicate the sampling distribution of a single-player Hedge algorithm over the joint action space, thereby inheriting its well-known regret guarantees. This has practical relevance for situations where communication is costly in terms of time or resources. Lastly, we tackle reward asymmetry by introducing `mHedge-B`, an extension of `mHedge-A`, which attains a regret bound of $\tilde{O}(M^2\sqrt{T})$. This result highlights how performance deteriorates as noise affects the observed rewards while still maintaining near optimal regret growth with respect to $T$.

## 2 Preliminary

**Single-Player Adversarial Full-Information Setting.** We first define the single-player adversarial full-information setting (sometimes loosely referred to as the bandit setting in this manuscript, though strictly speaking the feedback model below is full-information rather than bandit, since the learner observes the entire loss vector $\ell_t$ at the end of each round). Let $K$ be the number of arms identified with the set $[K] = \{1, 2, \ldots, K\}$ and let $T$ be the time horizon. The problem proceeds in rounds $t = 1, 2, \ldots, T$ as follows:

1. First, an adversary selects a loss vector $\ell_t \in [0,1]^K$, where $\ell_t(a)$ represents the loss for arm $a$ in round $t$. The adversary may choose these losses adaptively based on the learner's previous actions.

2. Then, without observing $\ell_t$, the learner selects an arm $a_t \in [K]$.

3. Finally, the learner observes the full loss vector $\ell_t$ and suffers the loss $\ell_t(a_t)$.

Let $\mathcal{H}_{t-1}$ denote all information available to the learner before round $t$, including the learner's past actions $a_1, \ldots, a_{t-1}$ and observed loss vectors $\ell_1, \ldots, \ell_{t-1}$. As mentioned above, since the adversary is adaptive, they may choose losses based on $\mathcal{H}_{t-1}$, and may also anticipate the learner's action $a_t$ in the current round.

The goal is to minimize the (expected) regret with respect to the best fixed arm in hindsight, defined below:

$$R_T = \mathbb{E}\left[\sum_{t=1}^{T} \ell_t(a_t)\right] - \min_{a \in [K]} \sum_{t=1}^{T} \ell_t(a), \tag{1}$$

where the expectation is taken over any randomness in the learner's algorithm (and over the adversary's randomness, if the adversary is randomized).

Note: The problem is often equivalently formulated with rewards $r_t(a) = 1 - \ell_t(a)$ instead of losses, in which case the goal is to maximize rewards and the regret definition changes accordingly.

**Oblivious vs Adaptive Adversary** It's worth noting that various characterizations of adversarial knowledge appear in the literature (refer to Related Works above), but this distinction is largely theoretical in our scenario. Under our approach of adversarial bandits, the essence of the problem lies in preparing for worst-case scenarios rather than average-case performance, as the adversarial framework makes no probabilistic assumptions about the environment. As such, we will assume below that our adversaries are adaptive, and we must deal with the omniscient adversary's decisions as needed. This also ensures that our results are generalizable towards all adversaries.

**Multi-Player Full-Information Adversarial Setting** We first define the multiplayer *full-information* adversarial setting. We use this model as a baseline before introducing asymmetric observations later. There are $M$ players, and player $m \in [M]$ has an action set $\mathcal{A}_m$. For now, assume $\mathcal{A}_m = [K]$ for all $m$, so the joint action space is

$$\mathcal{A} = \mathcal{A}_1 \times \cdots \times \mathcal{A}_M = [K]^M.$$

A joint action is denoted by $\mathbf{a} = (a_1, \ldots, a_M) \in \mathcal{A}$, where $a_m$ is the action chosen by player $m$.

The interaction proceeds over rounds $t = 1, 2, \ldots, T$ as follows:

1. The adversary chooses a loss function
$$\ell_t : \mathcal{A} \to [0, 1],$$
   where $\ell_t(\mathbf{a})$ is the loss incurred if the players jointly choose $\mathbf{a}$. Equivalently, one may view $\ell_t$ as a vector in $[0, 1]^{\mathcal{A}}$ indexed by joint actions.

2. Without observing $\ell_t$, each player $m$ selects an action $a_{t,m} \in \mathcal{A}_m$. Together these choices form the joint action
$$\mathbf{a}_t = (a_{t,1}, \ldots, a_{t,M}) \in \mathcal{A}.$$

3. After acting, every player observes the *entire* loss function $\ell_t$ (that is, the loss of every joint action), and the team incurs loss $\ell_t(\mathbf{a}_t)$. In particular, each player can compute the loss $\ell_t(\mathbf{a}_t)$ incurred by the team, as well as the counterfactual loss of any other joint action.

The players may agree on a joint strategy before learning begins—at which point they know the number of players $M$, each player's action set size $K$, and the time horizon $T$—but they cannot communicate during the learning process. The regret is measured against the best fixed joint action in hindsight:

$$R_T = \sum_{t=1}^{T} \ell_t(\mathbf{a}_t) - \min_{\mathbf{a} \in \mathcal{A}} \sum_{t=1}^{T} \ell_t(\mathbf{a}).$$

More generally, the players may have different numbers of actions; the above definitions extend immediately by letting each $\mathcal{A}_m$ be an arbitrary finite action set (so that $\mathcal{A}$ is the corresponding product space).

## 3 A Linear-Regret Lower Bound for Simultaneous Action Selection

In Chang et al. (2022); Chang & Lu (2024; 2025); Chang & Karthik (2025) they were able to prove near optimal regret bounds for the multiplayer information asymmetry bandit problem in the stochastic setting. In this section we construct an adaptive adversary (the skull adversary) that causes the learner to experience linear regret in the adversarial setting. We note that this impossibility is *not* a straightforward consequence of the adversarial setting: in single-player adversarial bandits, sublinear regret is achievable (e.g., $O(\sqrt{T \log K})$ via Hedge). The linear regret here arises specifically from the interaction between adversarial losses and the *lack of communication* among players. The key insight is that without communication, players can only implement product distributions over the joint action space, while the adversary can exploit the gap between the $(K-1)M$-dimensional space of product distributions and the full $K^M$-dimensional simplex over joint actions. To our knowledge, this lower bound technique is novel; while prior works such as Bubeck et al. (2020); Alatur et al. (2020) have studied adversarial multiplayer bandits, they consider collision-based models rather

than the information asymmetry framework studied here. Recall the multiplayer adversarial bandit setting defined above, where players may employ stochastic strategies, and decide their marginal arms simultaneously. As mentioned above, we assume each player also has exactly $K$ arms.

Since the players are not allowed to communicate, they can only decide on a random policy on their own action space (rather than the joint actions space). Therefore, the players' strategy space has $(K-1)M$ degrees of freedom, as each player can independently assign probabilities to $K-1$ of their actions (with the probability of the $K$-th action being determined by the constraint that probabilities sum to 1). However, the joint action space contains $K^M$ possible action combinations. This dimensional discrepancy implies that there exist numerous distributions over the joint action space that cannot be realized by any combination of independent player strategies, which allows any space-exploitative adversary to force higher regret bounds, assuming no communication is allowed.

Using that bit of wisdom, we now demonstrate that an adaptive adversary can force the players to incur linear regret in this setting.

**Theorem 1 (Skull Adversary Theorem)** *In a two-player adversarial bandit game where each player has two actions, there exists an adaptive adversary strategy (the Skull Adversary) that forces players to incur $\Omega(T)$ regret.*

**Proof:** Consider a scenario with two players ($M = 2$), each having two actions ($K = 2$). Let the joint action space be denoted by $a_{1,1}, a_{1,2}, a_{2,1}, a_{2,2}$, where $a_{i,j}$ represents player 1 selecting action $i$ and player 2 selecting action $j$. We define the Skull Adversary strategy as follows. At each round $t$, the adversary assigns losses $\ell_t^S$ (we use the superscript $S$ for "Skull") as follows:

1. Assign loss $\ell_t^S(a_{1,2}) = \ell_t^S(a_{2,1}) = 1$ for all rounds.

2. Let $p_t(a_{i,j})$ denote the probability of joint action $a_{i,j}$ under the players' strategy profile at round $t$. For the remaining two actions, assign:

   (a) $\ell_t^S(a_{i,j}) = 0$ to the joint action $a_{i,j} \in \{a_{1,1}, a_{2,2}\}$ with $\min\{p_t(a_{1,1}), p_t(a_{2,2})\}$
   (b) $\ell_t^S(a_{i,j}) = 1/2$ to the other joint action in $\{a_{1,1}, a_{2,2}\}$
   (c) In the case where $p_t(a_{1,1}) = p_t(a_{2,2})$, assign $\ell_t^S(a_{1,1}) = 1/2$ and $\ell_t^S(a_{2,2}) = 0$

Let $p_{i,j}$ and $\ell_{i,j}$ be the probability and resulting loss, respectively, of player 1 picking arm $i$ and player 2 picking arm $j$. Writing the joint action as a pair (player 1's arm, player 2's arm), the resulting loss table is

| (P1, P2) | $j = 1$ | $j = 2$ |
|----------|---------|---------|
| $i = 1$  | $x$     | $1$     |
| $i = 2$  | $1$     | $0.5 - x$ |

where $x = 0.5 \times \mathbf{1}(p_{1,1} \geq p_{2,2})$ with $\mathbf{1}$ being the commonly defined indicator function.

To show that the Skull Adversary ensures linear regret in $T$, let us first consider a lower bound for the loss of the overall best joint arm (in hindsight). Note that arms $a_{1,2}$ and $a_{2,1}$ both induce a total loss of $T$. For the remaining arms, note that

$$\sum_{t=1}^{T} \ell_t^S(a_{1,1}) + \sum_{t=1}^{T} \ell_t^S(a_{2,2}) = \sum_{t=1}^{T}(\ell_t^S(a_{1,1}) + \ell_t^S(a_{2,2})) = \sum_{t=1}^{T} \frac{1}{2} = T/2,$$

which implies that either $\sum_{t=1}^{T} \ell_t^S(a_{1,1})$ or $\sum_{t=1}^{T} \ell_t^S(a_{2,2})$ must be at most $T/4$ since both are positive and sum up to $T/2$.

Now we consider the optimized loss of any non-communicative strategy. WLOG, assume that $\ell^S(a_{1,1}) = 1/2$ (i.e., $x = 1/2$, which occurs when $p_{1,1} \geq p_{2,2}$). Let $q = p_{1,\cdot}$ and $r = p_{\cdot,1}$. Writing $\langle \ell, p \rangle_t := \sum_{\boldsymbol{a} \in \mathcal{A}} p_t(\boldsymbol{a}) \, \ell_t(\boldsymbol{a})$ for the expected loss under the players' joint product distribution at round $t$, the loss in round $t$ is then

$$
\begin{aligned}
\langle \ell, p \rangle_t &= \frac{1}{2} \times p_{1,1} + 1 \times p_{1,2} + 1 \times p_{2,1} + 0 \times p_{2,2} \\
&= \frac{1}{2} qr + q(1-r) + r(1-q) \\
&= -\frac{3}{2} qr + q + r \\
&= \frac{4 - (3q-2)(3r-2)}{6}.
\end{aligned}
$$

We cannot have both $q < 1/2$ and $r < 1/2$ since that would mean $qr < (1-q)(1-r) \iff p_{1,1} < p_{2,2}$. Thus, WLOG assume $q \geq 1/2$. Since $3q - 2 \in [-1/2, 1]$ and $3r - 2 \in [-2, 1]$, we have $(3q-2)(3r-2) \leq 1$. Thus,

$$
\frac{4 - (3q-2)(3r-2)}{6} \geq \frac{4-1}{6} = \frac{1}{2}
$$

so the loss for each round is at least $\frac{1}{2}$. Hence, for any given simultaneous policy,

$$
R_T = \sum_{t=1}^T \langle \ell_t^S, p_t \rangle - \min_{\boldsymbol{a} \in \mathcal{A}} \sum_{t=1}^T \ell_t^S(\boldsymbol{a}) \geq \frac{T}{2} - \frac{T}{4} = \frac{T}{4}, \tag{2}
$$

which implies that the Skull Adversary must always induce linear regret.

$\square$

It suffices to generalize the Skull Adversary to the general case, which will be done below.

**Theorem 2 (Generalized Skull Adversary Theorem)** *Let $M \geq 2$ be the number of players and $K \geq 2$ be the number of actions per player. For any set of independent stochastic strategies $\{\mathcal{P}_i \in \Delta(K)\}_{i=1}^M$, there exists an adaptive adversarial strategy that forces the players to incur $\Omega(T)$ regret.*

**Proof:** We proceed by constructing an adversarial strategy that extends from Theorem 1.

Define the loss function $\ell^* : [K]^M \to [0,1]$ as:

$$
\ell^*(a_1, a_2, \ldots, a_M) = \begin{cases} \ell_t^S(a_1, a_2) & \text{if } (a_3, \ldots, a_M) = (1, \ldots, 1) \text{ and } (a_1, a_2) \in \{1, 2\}^2 \\ 1 & \text{otherwise} \end{cases}, \tag{3}
$$

where $\ell_t^S$ is the Skull Adversary strategy established in Theorem 1. This forces players 1 and 2 to only pick arms 1 and 2, and players 3 onwards to pick arm 1, as any other set of actions would lead to loss 1. The problem of optimizing loss then reduces to the two-player, two-armed problem shown in Theorem 1, yielding $\Omega(T)$ regret. $\square$

**Remark 3** *Note that the $\Omega(T)$ lower bound in Theorem 2 is independent of both $K$ (the number of arms per player) and $M$ (the number of players). This is perhaps surprising: adding more players or more arms does not help overcome the fundamental impossibility of achieving sublinear regret without communication. Intuitively, the adversary can always reduce the problem to the 2-player, 2-arm case by assigning loss 1 to all other joint actions. Furthermore, the adversary constructed above requires knowledge of the players' strategy (i.e., their action probabilities) at each round; this is consistent with our adaptive adversary model.*

## 4 Sequential Action Selection under Full Information

As no policy can achieve sublinear regret in the simultaneous-pull setting without communication (Section 3), we now investigate an alternative formulation using successive pulls. In this model, players act in a

---

**Algorithm 1** `mHedge-A`

---

1: **Input:** learning rate $\eta > 0$
2: **Initialize:** for each player $i \in [M]$, set

$$L_0^{(i)}(\mathbf{a}) \leftarrow 0 \qquad \forall \mathbf{a} \in [K]^M.$$

3: **for** $t = 1$ to $T$ **do**
4:      Initialize the played prefix $\mathbf{a}_{t,1:0} \leftarrow \emptyset$
5:      **for** $i = 1$ to $M$ **do**
6:          Define the prefix-consistent set $\mathcal{A}_i(\mathbf{a}_{t,1:i-1}) := \left\{ \mathbf{a} \in [K]^M : \mathbf{a}_{1:i-1} = \mathbf{a}_{t,1:i-1} \right\}.$
7:          Player $i$ forms the distribution $p_t^{(i)}(\mathbf{a} \mid \mathbf{a}_{t,1:i-1}) = \dfrac{\exp\!\left(-\eta L_{t-1}^{(i)}(\mathbf{a})\right)}{\sum\limits_{\mathbf{b} \in \mathcal{A}_i(\mathbf{a}_{t,1:i-1})} \exp\!\left(-\eta L_{t-1}^{(i)}(\mathbf{b})\right)} \qquad \forall \mathbf{a} \in \mathcal{A}_i(\mathbf{a}_{t,1:i-1}).$
8:          Player $i$ samples a joint action $\mathbf{a}_t^{(i)} \sim p_t^{(i)}(\cdot \mid \mathbf{a}_{t,1:i-1}).$
9:          Player $i$ plays only the $i$-th component of this sampled joint action: $a_{t,i} \leftarrow \mathbf{a}_t^{(i)}[i].$
10:        Update the played prefix: $\mathbf{a}_{t,1:i} \leftarrow (a_{t,1}, \ldots, a_{t,i}).$
11:      **end for**
12:      The realized joint action is

$$\mathbf{a}_t \leftarrow (a_{t,1}, \ldots, a_{t,M}).$$

13:      Each player $i$ observes the full loss vector $\ell_t^{(i)} \in [0,1]^{K^M}$ (identical across players: $\ell_t^{(i)} = \ell_t$ for all $i$).
14:      **for** $i = 1$ to $M$ **do**
15:          Update cumulative losses:

$$L_t^{(i)}(\mathbf{a}) \leftarrow L_{t-1}^{(i)}(\mathbf{a}) + \ell_t^{(i)}(\mathbf{a}) \qquad \forall \mathbf{a} \in [K]^M.$$

16:      **end for**
17: **end for**

---

predetermined sequence where each player observes all preceding players' actions before making their own decision. This one-directional information flow effectively addresses the communication constraints that led to our impossibility result. We consider two sub-problems: Problem A, where all players observe the same (true) loss function (no information asymmetry), and Problem B, where each player observes a noisy version of the loss (reward asymmetry). We do not study action asymmetry in this sequential setting, as the successive observation of prior players' actions already resolves it; the remaining challenge is reward asymmetry. We note that throughout this section, we assume all players share the same number of arms $K$; this is primarily for notational convenience, and the results extend straightforwardly to the case of heterogeneous action sets.

### 4.1 Problem A: No Information Asymmetry

In this section, we propose `mHedge-A`, motivated by the single player hedge in Freund & Schapire (1997), where the $i$-th player will observe the previous $i - 1$-players actions $(a_1, \ldots, a_{i-1})$, and apply hedge on all the joint actions such that the first $i - 1$ components of that action match $(a_1, \ldots, a_{i-1})$. In other words they are applying a distribution over $K^{M-i+1}$ actions. Note that despite sampling across the joint action space, they can only pull their corresponding action. With an application of Bayes' rule, we are able to show that the resulting probability of sampling a joint action $\boldsymbol{a}$ is actually equal to the probability that a single player applying hedge across the entire joint action space would pick $\boldsymbol{a}$. Thus we are able to apply their well known regret bound. The pseudocode is given in Algorithm 1.

We now present the regret bound for `mHedge-A`.

**Theorem 4** *With learning rate $\eta = \sqrt{\frac{M \log K}{T}}$, the regret of `mHedge-A` (Algorithm 1) satisfies*

$$R_T \leq 2\sqrt{TM \log(K)}. \tag{4}$$

**Proof:** Given a joint action $(a_1, \ldots, a_M)$, consider the probability that it is pulled. By Bayes' rule, this is given by

$$\mathbb{P}(\boldsymbol{a} = (a_1, \ldots, a_M)) = \prod_{i=1}^{M} P\left(\boldsymbol{a}[i] = a_i \,\middle|\, \bigwedge_{j=1}^{i-1} \boldsymbol{a}[j] = a_j\right).$$

(For clarity, summing over some condition like $\boldsymbol{a}[1] = a_i$ is to sum over all $\boldsymbol{a} \in [K]^M$ that satisfy the condition. List slicing of the form $[a : b]$ denotes the $a$ through $(b-1)$st elements.)

Note that for player 1, the probability that they pull action $a_1$ is

$$\mathbb{P}(\boldsymbol{a}[1] = a_1) = \frac{\sum\limits_{\boldsymbol{a}[1]=a_i} \exp(-\eta L_{t-1}(\boldsymbol{a}))}{\sum_{\boldsymbol{a}} \exp(-\eta L_{t-1}(\boldsymbol{a}))}.$$

By induction, we can show that for player $i$, the probability that they pull action $a_i$ given the previous $i-1$ actions is

$$\mathbb{P}(\boldsymbol{a}[i] = a_i | \boldsymbol{a}[1] = a_1, \ldots, \boldsymbol{a}[i-1] = a_{i-1}) = \frac{\sum\limits_{\boldsymbol{a}[1:i+1]=(a_1,a_2,\ldots,a_i)} \exp(-\eta L_{t-1}(\boldsymbol{a}))}{\sum\limits_{\boldsymbol{a}[1:i]=(a_1,a_2,\ldots,a_{i-1})} \exp(-\eta L_{t-1}(\boldsymbol{a}))}. \tag{5}$$

Note that in the numerator we are summing across actions where the first $i$ components align with $(a_1, \ldots, a_M)$, while in the denominator, we are summing across actions where only the first $i-1$ components align with $(a_1, \ldots, a_M)$. Therefore, based on Bayes' rule above, we have

$$\begin{aligned}
\mathbb{P}(\boldsymbol{a} = (a_1, \ldots, a_M)) &= \prod_{i=1}^{M} \frac{\sum\limits_{\boldsymbol{a}[1:i+1]=(a_1,a_2,\ldots,a_i)} \exp(-\eta L_{t-1}(\boldsymbol{a}))}{\sum\limits_{\boldsymbol{a}[1:i]=(a_1,a_2,\ldots,a_{i-1})} \exp(-\eta L_{t-1}(\boldsymbol{a}))} \\
&= \frac{\exp(-\eta L_{t-1}((a_1, \ldots, a_M)))}{\sum_{\boldsymbol{a}} \exp(-\eta L_{t-1}(\boldsymbol{a}))}
\end{aligned}$$

where the last equality uses the fact that the product telescopes.

However, note that this would be the probability that $(a_1, \ldots, a_M)$ is sampled if there were only one player who played according to Hedge. Therefore we can apply the results from the regret analysis for the single-player Hedge algorithm in Luo (2022) to obtain the desired regret bound. In particular,

$$R_T \leq \frac{\log(K^M)}{\eta} + T\eta \leq 2\sqrt{TM \log(K)},$$

with $\eta = \sqrt{\frac{M \log K}{T}}$. $\qquad \square$

**Remark 5** *The $O(\sqrt{TM \log K})$ bound of Theorem 4 is near-optimal: since the joint action space has $K^M$ actions, the single-player Hedge lower bound of $\Omega(\sqrt{T \log(K^M)}) = \Omega(\sqrt{TM \log K})$ from Orabona & Pál (2016) applies directly, as our reduction shows the multiplayer problem is at least as hard as single-player Hedge over $K^M$ actions. Thus the bound is tight up to constant factors.*

### 4.2  Problem B: Information Asymmetry in Rewards

In this section, we propose `mHedge-B` which deals with the successive pulls in the case where the players receive an adversarial loss plus some stochastic subgaussian noise with 0 mean. Formally, at each round $t$, player $i$ observes the modified loss vector $\ell_t + X_t^i$, where $X_t^i \in \mathbb{R}^{K^M}$ has independent, 1-sub-Gaussian coordinates with zero mean. The noise vectors $X_t^i$ are independent across both rounds $t$ and players $i$, and

are not under the adversary's control. Each player knows the sub-Gaussian parameter $\sigma = 1$. Remarkably, we can apply the same principle used in `mHedge-A` in the previous section to obtain a nearly optimal regret bound for this setting as well. More specifically, the $i$-th player will observe the previous $(i-1)$-players actions $(a_1, \ldots, a_{i-1})$, and apply hedge on all the joint actions such that the first $(i-1)$ components of that action match $(a_1, \ldots, a_{i-1})$. However, when we apply Bayes rule, the stochastic losses prevent the probabilities of sampling a joint action from being equal to the standard hedge algorithm. However, we can upper bound these probabilities via using the standard concentration properties of sub-Gaussian distributions, characterized by the following well-known "Corollary 5.5" inequality.

**Lemma 6 (Corollary 5.5 of Lattimore & Szepesvári (2020))** *Assume that $X_i - \mu$ are independent, $\sigma$-subgaussian random variables. Then for any $\varepsilon \geq 0$,*

$$\mathbb{P}(\hat{\mu} \geq \mu + \varepsilon) \leq \exp\left(-\frac{n\varepsilon^2}{2\sigma^2}\right)$$

$$and \quad \mathbb{P}(\hat{\mu} \leq \mu - \varepsilon) \leq \exp\left(-\frac{n\varepsilon^2}{2\sigma^2}\right)$$

*where $\hat{\mu} = \frac{1}{n}\sum_{t=1}^{n} X_t$.*

Since the noise is 1-sub-Gaussian, we can construct the following interval, which contains the empirical mean of the noise with high probability

$$G_{\boldsymbol{a}}(t) = \left\{ \widehat{X}_{\boldsymbol{a}}(t) \in \left(-\sqrt{\frac{4\log(TK^M)}{t-1}}, \sqrt{\frac{4\log(TK^M)}{t-1}}\right) \right\}$$

Let the "good event" be defined as

$$G = \bigcap_{t=1}^{T} \bigcap_{\boldsymbol{a}} G_{\boldsymbol{a}}(t). \tag{6}$$

the extra $K^M$ factor will allow us to remove the exponential dependence on the action space in the regret bound. We have the following lemma, which upper bounds the probability of the complement of $G$.

**Lemma 7** *We have*

$$\mathbb{P}(G^c) \leq \frac{1}{T}$$

**Proof:** Using DeMorgan's laws and the probability union bound we have

$$\mathbb{P}(G^c) = P\left(\bigcup_{t=1}^{T} \bigcup_{\boldsymbol{a}} G_{\boldsymbol{a}}(t)^c\right)$$

$$\leq \sum_{t=1}^{T} \sum_{\boldsymbol{a}} \mathbb{P}(G_{\boldsymbol{a}}(t)^c)$$

$$\leq \sum_{t=1}^{T} \sum_{\boldsymbol{a}} \exp\left(-\frac{t-1}{2}\left(\sqrt{\frac{4\log(K^M T)}{t-1}}\right)^2\right)$$

$$\leq \sum_{t=1}^{T} \sum_{\boldsymbol{a}} \frac{1}{(K^M T)^2}$$

$$\leq \frac{1}{T}$$

where in the second inequality, we use Lemma 6 with $n = t - 1$, since at round $t$, we have $t - 1$ samples of the stochastic noise. $\qquad \square$

Players take turns pulling successively and condition their pulls on the previous players' pulls.

---

**Algorithm 2** `mHedge-B`

---

1: **Input:** learning rate $\eta > 0$
2: **Initialize:** for each player $i \in [M]$, set $L_0^{(i)}(\mathbf{a}) \leftarrow 0$ for all $\mathbf{a} \in [K]^M$
3: **for** $t = 1$ to $T$ **do**
4:     Initialize the played prefix $\mathbf{a}_{t,1:0} \leftarrow \emptyset$
5:     **for** $i = 1$ to $M$ **do**
6:         Define the prefix-consistent set $\mathcal{A}_i(\mathbf{a}_{t,1:i-1}) := \left\{ \mathbf{a} \in [K]^M : a_j = a_{t,j} \text{ for all } j < i \right\}$
7:         Player $i$ forms the distribution $p_t^{(i)}(\mathbf{a} \mid \mathbf{a}_{t,1:i-1}) = \dfrac{\exp\left(-\eta L_{t-1}^{(i)}(\mathbf{a})\right)}{\displaystyle\sum_{\mathbf{b} \in \mathcal{A}_i(\mathbf{a}_{t,1:i-1})} \exp\left(-\eta L_{t-1}^{(i)}(\mathbf{b})\right)}$ for all $\mathbf{a} \in$

  $\mathcal{A}_i(\mathbf{a}_{t,1:i-1})$
8:         Player $i$ samples a joint action $\mathbf{a}_t^{(i)} \sim p_t^{(i)}(\cdot \mid \mathbf{a}_{t,1:i-1})$
9:         Player $i$ plays only the $i$-th component: $a_{t,i} \leftarrow \mathbf{a}_t^{(i)}[i]$
10:         Update the played prefix: $\mathbf{a}_{t,1:i} \leftarrow (a_{t,1}, \ldots, a_{t,i})$
11:     **end for**
12:     The realized joint action is $\mathbf{a}_t \leftarrow (a_{t,1}, \ldots, a_{t,M})$
13:     Each player $i$ observes the noisy full-information loss vector $\ell_t^{(i)} := \ell_t + X_t^{(i)} \in \mathbb{R}^{K^M}$, where each coordinate $X_t^{(i)}(\mathbf{a})$ is 1-sub-Gaussian
14:     **for** $i = 1$ to $M$ **do**
15:         Update cumulative losses: $L_t^{(i)}(\mathbf{a}) \leftarrow L_{t-1}^{(i)}(\mathbf{a}) + \ell_t^{(i)}(\mathbf{a})$ for all $\mathbf{a} \in [K]^M$
16:     **end for**
17: **end for**

---

**Theorem 8** *With learning rate $\eta = \sqrt{\dfrac{M \log K}{T}}$, `mHedge-B` (Algorithm 2) gives a regret bound of*

$$R_T \leq 4M^2 \ln K \sqrt{2T \log T} + 1. \tag{7}$$

Comparing this to the regret bound in Theorem 4, the bound is the same order in $T$, but contains an extra factor of $M$ as a result of the union bound over the players' independent noise realizations.

**Proof:**   Note that we can decompose the regret as

$$R_T = \sum_{t=1}^T \mathbb{E}[\ell_t(a_t)] - \min_{a \in [K]} \sum_{t=1}^T \ell_t(a) = \sum_{t=1}^T \mathbb{E}[\ell_t(a_t) - \ell_t(a^*)]$$

where $a^*$ is the best arm in hindsight and the expectation is over the policy as well as the stochastic noise. Using the total law of expectation we obtain

$$R_T = \sum_{t=1}^T \mathbb{E}[\ell_t(a_t) - \ell_t(a^*)|G]\mathbb{P}(G) + \sum_{t=1}^T \mathbb{E}[\ell_t(a_t) - \ell_t(a^*)|G^c]\mathbb{P}(G^c)$$

$$\leq \sum_{t=1}^T \mathbb{E}[\ell_t(a_t) - \ell_t(a^*)|G] + T\mathbb{P}(G^c)$$

$$\leq \sum_{t=1}^T \mathbb{E}[\ell_t(a_t) - \ell_t(a^*)|G] + 1.$$

In the first inequality, we use the fact that the adversarial part of each loss is in $[0,1]$, and that the noise has averages out to 0. Therefore, the regret (under the condition of the complement of the good event) is upper bounded by $T$. The last inequality uses Lemma 7.

We now bound the regret under good event in equation (6). Given a joint action $(a_1, \ldots, a_M)$, we calculate the probability that it is pulled. By Bayes' rule, this is given by

$$\mathbb{P}(\boldsymbol{a} = (a_1, \ldots, a_M)) = \prod_{i=1}^{M} P\left(\boldsymbol{a}[i] = a_i \,\middle|\, \bigwedge_{j=1}^{i-1} \boldsymbol{a}[j] = a_j\right).$$

Note that for player 1, the probability that they pull action $a_1$ is

$$\mathbb{P}(\boldsymbol{a}[1] = a_1) = \frac{\sum\limits_{\boldsymbol{a}[1]=a_1} \exp\left(-\eta L_{t-1}^1(\boldsymbol{a})\right)}{\sum_{\boldsymbol{a}} \exp\left(-\eta L_{t-1}^1(\boldsymbol{a})\right)} \frac{\sum\limits_{\boldsymbol{a}[1]=a_1} \exp(-\eta L_{t-1}(\boldsymbol{a})) \exp\left(-\eta(t-1)\widehat{X}_{\boldsymbol{a}}^1(t-1)\right)}{\sum_{\boldsymbol{a}} \exp(-\eta L_{t-1}(\boldsymbol{a})) \exp\left(-\eta(t-1)\widehat{X}_{\boldsymbol{a}}^1(t-1)\right)},$$

where we have split the losses in the adversarial part as well as the stochastic part. Note the extra factor of $t-1$ since we use the sum of the noises. For player $i$, the probability that they pull action $a_i$ given the previous $i-1$ actions is

$$\mathbb{P}(\boldsymbol{a}[i] = a_i | \boldsymbol{a}[1] = a_1, \ldots, \boldsymbol{a}[i-1] = a_{i-1})$$
$$= \frac{\sum\limits_{\boldsymbol{a}[1:i+1]=(a_1,\ldots,a_i)} \exp\left(-\eta L_{t-1}^i(\boldsymbol{a})\right)}{\sum\limits_{\boldsymbol{a}[1:i]=(a_1,\ldots,a_{i-1})} \exp\left(-\eta L_{t-1}^i(\boldsymbol{a})\right)}$$
$$= \frac{\sum\limits_{\boldsymbol{a}[1:i+1]=(a_1,\ldots,a_i)} \exp(-\eta L_{t-1}(\boldsymbol{a}))}{\sum\limits_{\boldsymbol{a}[1:i]=(a_1,\ldots,a_{i-1})} \exp(-\eta L_{t-1}(\boldsymbol{a}))} \cdot \frac{\exp\left(-\eta(t-1)\widehat{X}_{\boldsymbol{a}}^i(t-1)\right)}{\exp\left(-\eta(t-1)\widehat{X}_{\boldsymbol{a}}^i(t-1)\right)}$$

Note that in the numerator we are summing across actions where the first $i$ components align with $(a_1, \ldots, a_M)$, while in the denominator, we are summing across actions where only the first $i-1$ components align with $(a_1, \ldots, a_M)$. Therefore, based on Bayes' rule above, we have

$$\mathbb{P}(\boldsymbol{a} = (a_1, \ldots, a_M)) = \prod_{i=1}^{M} \frac{\sum\limits_{\boldsymbol{a}[1:i+1]=(a_1,\ldots,a_i)} \exp(-\eta L_{t-1}(\boldsymbol{a}))}{\sum\limits_{\boldsymbol{a}[1:i]=(a_1,\ldots,a_{i-1})} \exp(-\eta L_{t-1}(\boldsymbol{a}))} \cdot \frac{\exp\left(-\eta(t-1)\widehat{X}_{\boldsymbol{a}}^i(t-1)\right)}{\exp(-\eta L_{t-1}(\boldsymbol{a})) \exp\left(-\eta(t-1)\widehat{X}_{\boldsymbol{a}}^i(t-1)\right)}$$

Note that even for actions that were not taken, we obtain a loss for that round. Therefore the "number of pulls" of each joint arm up to and not including time $t$ is $t-1$. Under the good event (6), we have $\widehat{X}_a \in (-\sqrt{2\log(TK^M)/(t-1)}, \sqrt{2\log(TK^M)/(t-1)})$. Therefore,

$$\mathbb{P}(\boldsymbol{a} = (a_1, \ldots, a_M)) \leq \prod_{i=1}^{M} \frac{\sum\limits_{\boldsymbol{a}[1:i+1]=(a_1,\ldots,a_i)} \exp(-\eta L_{t-1}(\boldsymbol{a}))}{\sum\limits_{\boldsymbol{a}[1:i]=(a_1,\ldots,a_{i-1})} \exp(-\eta L_{t-1}(\boldsymbol{a}))} \cdot \exp\left(2\eta\sqrt{2(t-1)\log(TK^M)}\right)$$
$$\leq \exp\left(2M\eta\sqrt{2(t-1)\log(TK^M)}\right) \cdot \frac{\exp(-\eta L_{t-1}((a_1, \ldots, a_M)))}{\sum_{\boldsymbol{a}} \exp(-\eta L_{t-1}(\boldsymbol{a}))}$$

where in the equality we used the fact that the adversarial part of the loss telescopes in the product. Note that for $x \in [0, 1]$, we have $e^x \leq 1 + 2x$. Therefore, the above can be bounded by

$$\mathbb{P}(\boldsymbol{a} = (a_1, \ldots, a_M)) \leq \left(1 + 4M\eta\sqrt{2(t-1)\log(TK^M)}\right) \cdot \frac{\exp(-\eta L_{t-1}((a_1, \ldots, a_M)))}{\sum_{\boldsymbol{a}} \cdot \exp(-\eta L_{t-1}(\boldsymbol{a}))}$$
$$\leq \left(1 + 4M\eta\sqrt{2T\log(TK^M)}\right) \cdot \frac{\exp(-\eta L_{t-1}((a_1, \ldots, a_M)))}{\sum_{\boldsymbol{a}} \exp(-\eta L_{t-1}(\boldsymbol{a}))}.$$

Note that the term in blue is the probability that $(a_1, \ldots, a_M)$ is sampled if there were only one player that played according to Hedge. Therefore, we can apply the regret analysis for the single-player Hedge algorithm in Luo (2022) and above, to deduce that

$$\sum_{t=1}^{T} \mathbb{E}[\ell_t(a_t) - \ell_t(a^*)|G] \leq \left(1 + 4M\eta\sqrt{2T\log(TK^M)}\right)\left(\frac{\log K^M}{\eta} + T\eta\right)$$

for the good event. Setting $\eta = \sqrt{\frac{M\log K}{T}}$ and adding back in the bad event term, we obtain the regret bound

$$R_T \leq 4M^2\ln K\sqrt{2T\log T} + 1.$$

Note that this requires $4M\eta\sqrt{2T\log(TK^M)} \leq 1$, which holds for sufficiently large $T$. For a fully rigorous treatment when $T$ is small, one can verify the bound directly. $\qquad\square$

## 5 Conclusions

We extended the multiplayer bandit framework of Chang et al. (2022) to adversarial environments. Our Skull Adversary construction shows that simultaneous action selection without communication inevitably leads to linear regret—contrasting sharply with the stochastic case. To overcome this, we introduced sequential action selection: `mHedge-A` achieves near-optimal $O(\sqrt{TM\log K})$ regret under identical losses, and `mHedge-B` achieves $\tilde{O}(M^2\sqrt{T})$ under reward asymmetry, showing that one-way communication suffices for sublinear regret.

**Practical implications.** Our model applies to any setting in which a centralized adversary can observe the players' strategies and adaptively choose losses to exploit them. A natural example arises in wireless networks with malicious jammers: multiple transmitters (players) each select a channel from their available set, and a jammer that monitors the transmitters' channel-selection distributions can concentrate its interference on the most likely joint channel allocation. In the simultaneous-selection regime, the Skull Adversary construction shows that such a jammer can guarantee linear disruption regardless of the transmitters' coordination strategy agreed upon before deployment. The sequential model we propose—where transmitters act in a fixed order, each observing the preceding selections—mirrors practical time-division or hierarchical access protocols and restores sublinear regret via `mHedge-A` and `mHedge-B`. More broadly, any distributed system where independent agents share a common environment and face a strategic opponent fits this framework: examples include multi-agent cybersecurity, where a coordinated attacker observes defenders' allocation of monitoring resources across network segments; multi-robot adversarial navigation, where an opponent can observe and obstruct the robots' planned paths; and competitive resource allocation in cloud computing, where a malicious scheduler can observe tenants' bidding strategies and degrade service accordingly. In each case, the key structural feature captured by our model is the gap between the $(K-1)M$-dimensional product strategy space available to non-communicating players and the full $K^M$-dimensional simplex that the adversary can exploit—and the sequential protocol's ability to close this gap through one-way observation.

**Limitations and future work.** Our model requires a fixed player ordering. Key open questions include whether weaker coordination (partial orderings, limited communication) suffices, whether matching lower bounds for Problem B can be established, and whether these results extend to bandit feedback.

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

## A    Additional Related Works

The adversarial bandits landscape has evolved substantially since the pioneering work of Auer et al. (1995), which introduced both the adversarial bandit problem and the seminal EXP3 algorithm (later refined in the journal version Auer et al. (2002)). This line continued with Audibert & Bubeck (2010), who refined these approaches through their Implicitly Normalized Forecaster, establishing tighter minimax optimal regret bounds that significantly improved performance. Bubeck & Slivkins (2012) further crafted sophisticated algorithms that leverage consistency conditions to achieve $\text{polylog}(T)$ regret in stochastic environments while maintaining $\tilde{O}(\sqrt{T})$ guarantees for adversarial settings. The challenge of developing algorithms that perform well without prior knowledge of the environment type was tackled by Seldin & Slivkins (2014) and later Auer & Chiang (2016), who enhanced EXP3 to create robust algorithms that adapt to the environment without requiring prior knowledge of whether it is stochastic or adversarial. This line of research culminated in the work of Zimmert & Seldin (2019), who developed theoretically optimal algorithms achieving the best possible regret bounds across arbitrary environments—including stochastic, adversarial, and corrupted regimes—effectively bridging the gap between these formulations.

Many other variations of adversarial models address practical challenges and connect to popular frameworks in the bandit literature. Contextual adversarial bandits, introduced by Auer (2002) and further developed by Beygelzimer et al. (2011) and Agarwal et al. (2014), incorporate side information available before decision-making, allowing algorithms to adapt their strategies based on observed contexts. This approach has been refined by Syrgkanis et al. (2016) with variance-based improvements. Combinatorial adversarial bandits Cesa-Bianchi & Lugosi (2012) significantly expand the action space by enabling learners to select subsets of base actions simultaneously, with Neu (2015) establishing tighter bounds for adversarial semi-bandit feedback; the closely related work of Kveton et al. (2015) studies the *stochastic* combinatorial semi-bandit setting rather than the adversarial one. The bandits with knapsacks framework, pioneered by Badanidiyuru et al. (2013), extended by Agrawal & Devanur (2016) and Rangi et al. (2018), and finally incorporated with adversarial patterns by Immorlica et al. (2022), incorporates resource constraints where actions consume from limited budgets, modeling real-world scenarios with capacity limitations or cost considerations. Dueling bandits Yue et al. (2012) represent a fundamental shift in feedback structure, focusing on preference-based learning where only relative comparisons between actions are observable rather than absolute rewards; Zoghi et al. (2014) and Komiyama et al. (2015) study the stochastic variant, and Saha et al. (2021) develops algorithms for the adversarial variant. Finally, corrupted adversarial models, as shown in Hajiesmaili et al. (2020) and Ma & Zhou (2023), consider corruption budgets for corrupted non-stochastic bandits to ensure robustness against adversaries modifying limited rewards or contexts.

Our work relates to the broad literature on multiplayer bandits, introduced by Awerbuch & Kleinberg (2008) for collective action selection. Subsequent work developed a range of approaches, including $\epsilon$-greedy variants Szorenyi et al. (2013); Jin et al. (2023), distributed methods such as gossip UCB Landgren et al. (2016); Martínez-Rubio et al. (2019), distinctive actions through time iterations Uchiya et al. (2010), and the leader-follower DPE1 framework Wang et al. (2020). Other coordination mechanisms include local observation constraints Cesa-Bianchi et al. (2016), communication over graph networks Xu & Klabjan (2024), asynchronous activation of player subsets Bonnefoi et al. (2017); Cesa-Bianchi et al. (2020), voting-based collective decisions Shahrampour et al. (2017), and communication among agents with personalized action sets Dubey et al. (2020). In most of these works, communication occurs during learning, allowing players to exchange feedback at some cost. Notable exceptions that study communication-free settings include Avner & Mannor (2014); Rosenski et al. (2016); Bistritz & Leshem (2018); Féraud et al. (2019), though these focus on stochastic rewards and collision models. By contrast, we allow communication only before learning begins, when players know only the environment size and the time horizon, and we consider adversarial losses.

Interest in multiplayer multi-armed bandits was initially driven by opportunistic spectrum sharing, with early works including Gai et al. (2012); Liu & Zhao (2010); Anandkumar et al. (2011). Related communication- and network-motivated formulations were studied in Maghsudi & Stańczak (2014); Korda et al. (2016); Shahrampour et al. (2017); Chakraborty et al. (2017). These papers either considered the centralized setting or the symmetric-user case, where all users share the same reward distributions, and typically assumed collisions: if two or more users choose the same arm, none receives a positive reward. The first general solution to this matching problem was given by Kalathil et al. (2014), achieving logarithmic-squared regret, which was improved to logarithmic regret in Nayyar et al. (2018) via posterior sampling. Because these methods relied on implicit and often costly communication, later work sought communication-free alternatives Avner & Mannor (2014); Rosenski et al. (2016); Bistritz & Leshem (2018); Féraud et al. (2019). Regret lower bounds were studied in Besson & Kaufmann (2018), and more recent work on decentralized learning for multiplayer matching bandits includes Wang et al. (2020); Shi et al. (2020).

Adversarial multiplayer bandits have also received growing attention. Collision-based adversarial models were studied in Alatur et al. (2020); Bubeck et al. (2020), with Alatur et al. (2020) in particular developing an adversarial collision-penalty framework motivated by radio networks. Related adversarial formulations have since been explored in radar networks Mahesh et al. (2024), CSMA networks Tong et al. (2021), and robustness-to-manipulation analyses through the notion of attackability Shi & Shen (2021). Distributed expert/online-learning settings with multiple players were also studied by Kanade et al. (2012). However, to the best of our knowledge, none of this literature studies the role of information asymmetry in the setting we consider, nor the more general case in which each agent may have its own action space.

