# OpenReview forum: "The Learnability of the Multiplayer Adversarial Bandit Problem"
_TMLR — Under review for TMLR_

### Review · Reviewer_kxoS · 2026-03-23

**Summary Of Contributions:**

In this work, the authors consider a problem of multiplayer multi-armed bandit as well as the multiplayer full information variant of the problem. In their problem setting, each player has access to their own set of arms, and thus, there is no risk of collision. Each player selects their own arm independently and without communication with the other players (besides some optional organization before the beginning of the game).

The authors show that against an adaptive adversary, meaning an adversary that has access to the losses of the previous rounds, if all the players select their own arm at the same time, then the adversary can force linear regret even when the players can observe the full loss vector at each round. (In a version of the problem where some pairs of actions are better than others).

Then, they consider another version of the problem where the players have to choose the arm they play one by one. They still cannot communicate with each other, but this allows a sort of contextual learning for the later players as they know what the previous players played.
The authors propose algorithms and upper bounds for the variants of this problem where all the players have access to the same information, and the version of the problem where the players have access to noisy observations of the joint loss.
In both cases, they show that it is possible to achieve $\sqrt T$ type regret, which highlights that the sequential arm pulling of the players is essential to by passing linear regret when playing against adaptive adversaries.

**Audience:**

Yes

**Audience Explanation:**

Understanding the limitations of learning is very important for the learning theory and the bandit community.

**Broader Impact Concerns:**

no problem.

**Claims And Evidence:**

Yes

**Claims Explanation:**

The paper is short but all the results contained detailed proofs.

I would point out that many small changes should be addressed before the paper is ready for publication, but the results seem interesting.

**Requested Changes:**

The paper would benefit from making the following changes:

In the literature review, the topic of multiplayer multi-armed bandits could be better covered. In particular, it is very common to consider a version of the problem where the players have to play on the same arms and so collisions can be a problem, especially in the adversarial setting, but they can also be used as a way to communicate (for example Boursier and Perchet 2018). As the setting where the players have their own arms is less common, it would be good to relate to the rest of the literature.

On page 2, you mix up using \citet and \citep in the citations towardds the bottom of the page.

On page 3, the paragraph title that is at the very bottom of the page should be put on the next page.

On page 4 (top of the page), the explanation of the multiplayer setting isn't very clear as people might be aware of a different variant of MPMAB. Also, calling this section bandit but stating that the players observe the full loss vector is confusing. At this point, the notion of asymmetry is also very difficult to understand and so it is difficult to follow what you are trying to explain until we reach the later sections.

The notion of variable action counts on page 4 is also unclear, and the notation used there (the inverted G) is not clear and not introduced.

Reward assymetry -> there is a typo on the word receives (was written recives)
The title of section 3 also seem confusing... on simultaneous what?

On page 5, the statement in the proof "... must be at most T/4" would benefit from being slightly more explained.

On page 6, towards the end of the proof, the indexing on the $\min_{a \in [a]}$ isn't super clear.
On the second proof on that page, the alignment of the first eqarray is off, and the last sentence of that proof "and we're done" isn't super formal.
The title of section 4 is also hard to read, and is confusing. In the subsequent algorithms, the player only observes its own part of the loss vector.
Also in Algorithm mHedge-A, what do you mean by "play p_t"? There is something incorrect about the way things are phrased.
(Same comments about the second algorithm).
In the proofs on pages 7-11, you have the same index on your sums at the numerator and denominator, which can be slightly confusing.

On page 10 in the large eqarray at the bottom consider putting spaces around $|$ to improve readability.

On page 11, add some space after "Therefore" at the top of the page so it is easier to read the long lines of math underneath.
The sentence "if there were only one player that played according to Hedge" is confusing. Are you trying to say that this reduces to a single player game where the player uses the Hedge algorithm?

---

> ### Author Response · Authors · 2026-03-26
> **response to reviewer**
>
> Thank you for the detailed and constructive review. We appreciate both your positive assessment of the results and your careful comments on the presentation.
>
> We have corrected the typographical and formatting issues you identified and updated the manuscript accordingly. This includes the noted typo in “receives,” the citation formatting inconsistencies, spacing and alignment issues in the proofs, and several readability improvements throughout.
>
> Regarding the literature review, we have expanded the discussion of multiplayer bandits, especially collision-based and adversarial settings, to better relate our work to the broader literature. In particular, we now discuss the following two papers, which are especially relevant:
> - Boursier and Perchet, *Multi-Player Bandits: The Adversarial Case*
> - Bubeck et al., *Non-stochastic Multi-player Multi-armed Bandits: Optimal Rate with Collision Information, Sublinear Without*
>
> We also revised the discussion on page 4 to make the multiplayer setting clearer. In particular, we now distinguish more carefully between the full-information adversarial setting we study there and other variants of multiplayer bandits, and we removed or clarified notation that was introduced too early. We also postponed the discussion of asymmetry until after the base setting is fully defined.
>
> For the statement in the proof that one quantity “must be at most \(T/4\),” the reason is simply that the two quantities sum to at most \(T/2\), so at least one of them must be at most \(T/4\); otherwise their sum would exceed \(T/2\). We have added this explanation in the manuscript.
>
> We have also revised the titles of Sections 3 and 4 to make their roles clearer. In the algorithm descriptions, we clarified the meaning of the action-selection step: player \(i\) samples a joint action consistent with the previously chosen prefix, but only plays the \(i\)-th component of that sampled joint action. We updated both algorithms to reflect this more accurately.
>
> Regarding the proof notation on page 6, the indices in the numerator and denominator are not intended to be the same. The numerator sums over joint actions matching the first \(i+1\) components of \(\bm a\), whereas the denominator sums over joint actions matching only the first \(i\) components. We agree that this was not sufficiently clear, and we have revised the presentation to make this distinction more explicit.
>
> Finally, on page 11, the intended meaning of the sentence about “only one player that played according to Hedge” is that the process can be viewed as equivalent to a single learner choosing the entire joint action directly, and therefore reducing to a standard single-player Hedge instance. We have rewritten this sentence to make that interpretation explicit and more formal.
>
> Thank you again for the thoughtful feedback. We believe these revisions have substantially improved the clarity of the paper.

---

### Review · Reviewer_gGpf · 2026-04-03

**Summary Of Contributions:**

This paper considers a multi-player adversarial multi-armed bandit problem, under various models of how information is shared before and after actions. In a setting where players cannot communicate during the game, observe the full loss function after each round and the adversary can adapt to the action probability vectors of the players, it is demonstrated that an adversarial strategy guaranteeing linear regret exists. In a setting where players can communicate during rounds, a variant of the Hedge strategy of Freund and Schapire (1997) can be developed which inherits the bound of Hedge since it can be shown that the multiple players combine in a way as if to play Hedge as a single-player over the joint action space. In a similar setting, but where players see (individually) noised versions of the loss, they can show that while the realised policy is not identical to the single-player Hedge, it is close with high probability, allowing an analysis tethered to the Hedge one.

A key strength of the paper is that it derives new theoretical results for an interesting set of problems. A key weakness however is that not much is done to articulate the benefit of these results to community or to reason about their limitations.

**Audience:**

Yes

**Audience Explanation:**

Some of the adversarial bandits and/or federated learning community may find these results useful.

**Broader Impact Concerns:**

I have no ethical concerns specific to this work. It is basically entirely theoretical and does not seem to unlock a new capability that would have an ethical impact beyond what the autonomous decision-making field already contributes.

**Claims And Evidence:**

Yes

**Claims Explanation:**

For the most part, yes the claims are clear and accurate - there are a couple of minor issues with the theory, see below, and parts where I think things could be clarified.

**Requested Changes:**

I have listed my concerns and suggestions below by section, using prefix CRITICAL: to highlight points which are critical, versus those which are more minor.

Abstract:
 - In general, I think this could be a bit clearer - it defers some of the explanation to Chang et al. rather than just describing the problem (which seems unecessary in the abstract). It would be useful to explain that the regret bounds are obtained for hedge-style algorithms, and that this is done by establishing links between the multi-player and single-player policies. Also, it should be clear that the players incur the regret, not the adversary.

1. Intro:
-	Para 1: Regret and (perhaps less critically) arms are undefined at the point of their discussion.
-	Para 2: i.i.d. observations mentioned here, but this paper is in an adversarial setting? Could you describe information asymmetry in more general terms here?
-	Para 3: First sentence doesn’t make sense, intersection of adversarial bandits with what?
-	Para 4 (Our Contribution): should add stochastic to first sentence?
-	CRITICAL: General: I don’t get a good sense of why this asymmetric information adversarial multi-player problem is interesting. There is a statement that adversarial bandits generally have been used in certain applications – albeit without references – and how distributed decision-making models robotics and resource allocation – again no references - but nothing to explain specifically where this model is useful. Later in the section there is some discussion of specific methods for spectrum sharing problems, but no clear link to the model in this paper.
-	Para 6 is a nice account but it’s not clear what it’s purpose is in the current paper – do these methods somehow influence the methods in this paper?
-	CRITICAL: Para 7: ‘in most of these works’ suggests that in some there is no communication during learning, suggesting they are relevant to the current work – can you highlight these?
2. Preliminary: Single-player adversarial bandits
-	Par 1, List item 1: $a \in [K]$ for clarity here.
-	Par 2, a more specific statement of what $\mathcal{A}_{t-1}$ contains should be given – e.g. any prior information? $a_t$’s as well as $l_t$’s? – to avoid confusion.
-	Par 2, it’s confusing to use $\mathcal{A}$ for histories here, because they are immediately action sets later in the section -  maybe use $\mathcal{H}$ for historic information?
2. Preliminary: Multi-player full-information adversarial setting
- Para 2, point 3. The players see the entire loss function, but do they know what loss the team incurs, and what specific action vector lead to that loss?
- Para 4: would the players know each others’ action sets in this case?
3. Lower Bounds
- Para 1: This sounds a lot like a potential contradiction, is it that they are studying a different environment or assumptions? Some extra clarity is needed here.
- CRITICAL: General: discuss whether this is expected or not, for regular adversarial bandits, does this happen, what about some of the settings mentioned in the related work, and does the proof of this use novel ideas or is it based on proofs from some of those other settings? This adversary also needs to know the players strategy (probabilities) in advance to construct the linear regret policy – is that standard?
- Proof of Theorem 1,
- what is $l_S$? Previously we have defined $l_t$, is $S$ a typo for $t$ or a subscript indicated the Skull strategy, if so, why in part 2 of its definition is it not $t$ dependent?
- Missing set brackets around $a_{1,1}, a_{2,2}$ on line (a)
- Page 5: ‘WLOG assume that $a_{1,1}=0.5$’ should this be $l(a_{1,1})$?
- MINOR BUT CRITICAL: P6: $3r-2$ should be in $[-2,-1/2]$ if r < ½?
- Page 6: the max over $[\mathbf{a}]$ uses undefined notation. Should be over $\mathcal{A}$?
- Theorem 2: I think it would be interesting to indicate explicitly in the theorem statement that this $\Omega(T)$ is independent of $K$ and $M$. Perhaps that is also worth some discussion afterward, is again is this surprising, that this adversary doesn’t make the regret any worse or better as more players or arms are added?
- CRITICAL: General: what are the practical implications of this result? In what settings might such an adversary realistically exist?
4. Sequential Action Selection
- Par 1: This first sentence is true for adaptive adversaries, I’d add that clarification.
- CRITICAL: General: The introduction mentions the prospect of action asymmetry as well as reward asymmetry, or the prospect of having both. Are those cases not worth studying here? Is the latter actually the case of Section 3. This could be more clearly explained here.
- CRITICAL: General: the bounds are described as nearly optimal in this section, but there is no reference to lower bounds for these problems (other than the one from Section 3 in a harder setting) – they should be referenced and articulated clearly to justify the claim of near-optimality.
- CRITICAL: Everything in this section seems to be for the case where all players have the same action set, is that correct? If so, it’s worth clarifying that, and discussing why/whether it is necessary.
4.1 Problem A:
- There is some mixing of notation where $a_{i:j}$ represent a sequence of elements of a vector and $a[i:j]$ do also. Is this necessary or could a consistent choice be adopted? Maybe I've missed a distinction between them?
- Alg 1 Line 12: does every player observe this action, or just the last one? Does it matter?
4.2 Problem B:
- Para 1: It seems worthwhile to make clearer the independence structure of the $X_t^i’$s here. Confirm if they are i.i.d. across rounds and players, free from the control of adversary and do the players know the sub-gaussian parameter is 1?
- CRITICAL: Theorem 6: The first term of the stated result seems to differ by a factor of 2\sqrt{K} from what is achieved at the last line of the proof.
- CRITICAL: Theorem 6: I think some more clarity is needed at the end regarding how the regret bound is obtained, when I try to follow the analysis, I get that the eta^2 term is linear in K, and I have some log(M) terms which I think are maybe bounded with M but I’m not sure.


Ultimately, the main concerns are any incorrect or ambiguous statements in the theory, and the justification of the importance of the work.

---

> ### Author Response · Authors · 2026-04-08
> **response to reviewer**
>
> We thank the reviewer for the thorough feedback. All changes have been implemented in the revised manuscript. Due to the character limit we provide a terse summary of our changes.
>
> ## Abstract & Introduction
>
> We rewrote the abstract to be self-contained: it now states the problem without deferring to Chang et al., names the Hedge connection explicitly, and clarifies that the *players* incur the regret. We also added concrete application references (multi-robot coordination, decentralized resource allocation) with citations, described information asymmetry in more general terms (removing "i.i.d."), defined regret and arms at first use, added "stochastic" to distinguish prior work, and highlighted which related works are communication-free.
>
> ## Section 2 (Preliminaries)
>
> We renamed $\mathcal{A}\_{t-1} \to \mathcal{H}\_{t-1}$ to avoid conflict with action sets, and explicitly listed its contents ($a_1,\ldots,a_{t-1}$ and $\ell_1,\ldots,\ell_{t-1}$). We also clarified that players observe the entire loss function (including counterfactuals) and that they know $M$, $K$, and $T$ at the agreement stage.
>
> ## Section 3 (Lower Bounds) — CRITICAL items
>
> **Why is this interesting / is linear regret expected?** We added a discussion explaining that linear regret is *not* expected in single-player adversarial bandits ($O(\sqrt{T\log K})$ via Hedge). The impossibility arises specifically from the interaction between adversarial losses and the restriction to product distributions imposed by the lack of communication. The technique is novel relative to prior collision-based models (Bubeck et al., 2020; Alatur et al., 2020).
>
> **Prior work contradiction?** Chang et al. study the *stochastic* setting, whereas our impossibility is specific to *adversarial* losses. The key coordination mechanism used in Chang et al. relies on arms having inherent stochastic qualities, which does not apply in the adversarial case.
>
> **$\Omega(T)$ independent of $K$ and $M$?** We added a remark explaining that the adversary can always reduce to the 2-player, 2-arm case by assigning loss 1 to all other joint actions.
>
> **Practical implications?** Any setting with a centralized adversary that can observe the players' strategies fits our model (e.g., malicious jammers in wireless networks). This is now discussed in the expanded conclusions.
>
> **Minor fixes:** We clarified the $\ell_S$ notation (superscript $S$ for "Skull"), fixed missing set brackets around $\{a_{1,1}, a_{2,2}\}$, corrected the typo "$a_{1,1}=0.5$" to "$\ell(a_{1,1})=1/2$", and fixed the $\min_{\mathbf{a}\in\mathcal{A}}$ notation. Regarding $3r-2 \in [-2,1]$: the proof is correct since WLOG $q\ge 1/2$ but $r\in[0,1]$, and $(3q-2)(3r-2)\le 1$ is tight at $(q,r)=(1/2,0)$.
>
> ## Section 4 (Sequential Action Selection) — CRITICAL items
>
> **Action vs. reward asymmetry:** We rewrote the section introduction. In the sequential model, action asymmetry is inherently resolved by observing predecessors' actions, so the remaining challenge is reward asymmetry (Problem B).
>
> **Near-optimality claims lack lower bounds:** We added a remark showing that our reduction establishes equivalence to single-player Hedge over $K^M$ actions, so the $\Omega(\sqrt{TM\log K})$ lower bound (Orabona & Pál, 2016) applies directly.
>
> **Same action set assumption:** We added a clarification that the equal-$K$ assumption is for notational convenience and that the results extend to heterogeneous action sets.
>
> **Theorem 6 discrepancy:** The learning rate was incorrectly written as $\eta=\sqrt{K\log M/T}$ (swapping $K$ and $M$). We corrected it to $\eta=\sqrt{M\log K/T}$, and the stated bound is now consistent. We also added a note on the condition required for the $e^x\le 1+2x$ approximation.
>
> **Independence of $X_t^i$:** We clarified that the noise vectors are independent across rounds and players, are not adversary-controlled, and have known sub-Gaussian parameter $\sigma=1$.
>
> **Other fixes:** We clarified the Algorithm 1 observation model and adopted consistent bracket notation with an explicit note.
>
> We hope these changes improve the exposition of our paper.

---

> > ### Comment · Reviewer_gGpf · 2026-04-08
> > **Discussion**
> >
> > Thank you for the prompt and systematic response to my comments. I appreciate these changes and think that the modifications made are appropriate to fix many concerns. I'd note a couple of outstanding items:
> >
> > 1. The updated $l_S$ notation now says that $l^S_t$ is used, but later in the proof it is still $l_S$.
> > 2. You mention that the centralized adversary setting is discussed in an expanded conclusions section, but if anything the conclusion is now shorter and just lists certain extensions rather than discuss them.

---

> > > ### Author Response · Authors · 2026-04-14
> > > **response to reviewer**
> > >
> > > We sincerely apologize for the oversight. We have modified the notations and the conclusions as such.

---

### Review · Reviewer_DFzA · 2026-05-06

**Summary Of Contributions:**

This paper studies an online learning problem with multiple players in the full information setting and adversarially chosen loss functions. (This problem is incorrectly termed “Multiplayer Adversarial Bandit” even though there is no bandit feedback.) The loss function is determined by the joint action of all players. The players are not allowed to communicate during the game, but are allowed to communicate prior to the game start, e.g., to agree on a strategy. Further, the players are assumed to be asyemmetric, in terms of available action set (action asymmetry) and possibly the perceived reward/loss/feedback (reward asymmetry).

The paper shows that a linear regret cannot be avoided against an adaptive adversary under simultanous arm pull (i.e., the classical model in online learning and bandits). Motivated by this, the paper proposes a model called successive pulls, where the players pull arms in a turn-based fashion. The paper proposes Hedge-style algorithms and reports corresponding regret bounds scaling as $\sqrt{T}$ (up to log factors).

**Strengths**

- The multiplayer variants of online learning and bandit problems are interesting and important as they align well with many realistic scenarios. Especially, consideration of heterogeneous action sets is well-motivated in practice.
- The related work section provides a good coverage for studies on multiplayer bandits in the stochastic setting, and to some extend in the adversarial setting. However, as discussed later, some works appear to be missing.
- The impossibility of sublinear regret against an adaptive adversary under simultaneous move is interesting, despite the simplicity of the construction.

**Weaknesses and Concerns**

**1. Technical Novelty.** Overall, the level of technical novelty appears limited. The proposed algorithms (mHedge-A/B) are largely straightforward adaptations of Hedge, and their regret analyses rely on standard arguments. In particular, the regret analysis on p. 9 is essentially a direct application of a Hoeffding-type inequality combined with a union bound, which is routine and does not constitute a substantial contribution on its own. As such, both the algorithmic ideas and their analyses read more like a technical exercise and do not justify technical contribution. In contrast, the lower bound analysis appears to contain some novel elements and deserves credit.

**2. Formulation of Problem B**. While incorporating asymmetry across players in terms of observed feedback (losses) is an interesting direction, the model introduced in Problem B appears to rely on stochastic noise in a way that is difficult to justify within the adversarial framework considered. This makes the formulation feel somewhat artificial. It also becomes apparent later that this assumption is introduced primarily to facilitate the analysis. Although removing such assumptions may raise questions about learnability, there may be alternative ways to preserve the adversarial nature of the losses without introducing arbitrarily differing feedback across players. As it stands, the assumptions underlying Problem B seem contrived and tailored to yield tractable analysis.

**3. Feasibility of the Successive Arm Pull Model.** How practical is the considered successive arm pull model? Its implementation aspects are not fully clear (even at the pseudocode level), and unless I am missing something, it silently assumes subdivision of each time slot.

**4. Presentational Issues.** One key concern relates to issues in presentation. The manuscript appears careless in its use of terminology.

- The feedback model under consideration corresponds to the full-information setting and should therefore not be described as “bandit.” This is not merely a pedantic point: the misuse of terminology can confuse readers and complicate comparisons with related work. For instance, Section 2 is titled “Single-player adversarial bandits,” yet the description clearly aligns with a full-information setting.
- The paper lacks precision in its notation and definitions. For example, the regret definition appears incomplete, as it does not account for the expectation over the algorithm’s possible randomization. Additionally, both $\text{Reg}_T$ and $R_T$ are used to denote regret, which introduces unnecessary inconsistency.
- The paper repeatedly alternates between “loss” and “reward” terminology, introducing further confusion. Although the technical sections eventually adopt the loss-based formulation, references to rewards continue to appear, leading to inconsistency.
- More broadly, the problem formulation occupies an unnecessarily large portion of the manuscript (approximately 1.5 pages). With improved organization and the removal of redundant or standard material, this could likely be condensed to about half that length without loss of clarity.


**Detailed Feedback**
- Both “multi-player” and “multiplayer” were used.
- The notation $<\ell,p>$ is not defined, even though a reader may guess what it is.
- Both $\text{Reg}_T$ and $R_T$ are used to denote the regret.
- For generality, I suggest $[0,1]^{\mathcal A}$ instead of $[0,1]^{K^M}$.
- Below p. 4, you repeat the definition of $\mathcal A$, which is redundant.
- In Section 4: Shouldn’t it be “(no reward asymmetry)” instead of “(no information asymmetry)”?
- In the context of Theorem 1, the meaning of $a_{i,j}$ is not clear. Does it refer to the action player $i$ thinks that player $j$ has chosen? If correct, then this setting has nothing to do with the setting and formulation in the preceding section. And if this is correct, the assertion of the theorem is rather trivial.
- On page 1, multiplayer adversarial bandit is abbreviated as MAB, which is not a good idea, since the acronym is readily widely used for "Multi-Armed Bandit". Also, the paper uses MAB in later places to denote multiarmed bandit. At a later stage, MMAB is used, without being defined.
- In terms of technical presentation, it would make more sense to present the two variants (successive and simultaneous pulls) in a section related to ‘the setting’, and then proceeding to presenting the (possibility/impossibility) results.
- Arguably, in the context of online learning and bandits, the term ‘Information Asymmetry’ could be vague, as information may be broadly interpreted (e.g., actions of other players, perceived feedback).
- Note that the regret bounds in Theorems 3 & 4 are only valid for a particular learning rate $\eta$, which must be mentioned in their corresponding statements.

**Related Work**

There are some imprecise pointers in the related work discussion.
- EXP3 was introduced and analyzed in (Auer et al., 1995). The statement that “Auer et al. (2002) developed the seminal EXP3 algorithm …” is incorrect. The bandit setting (and EXP3) were considered already in Auer et al. (1995).
- Concerning (Bubeck & Slivkins, 2012)): under the adversarial setting, their regret performance is $\tilde O(\sqrt{T})$ not $\tilde O(T)$.
- Note that Kveton et al. (2015) study *stochastic* combinatorial bandits, and hence is not relevant in the context of adversarial combinatorial bandits.
- The Regarding dueling bandits, the statement “developing efficient algorithms for both stochastic and adversarial variants” is misleading. Unless the reader is an expert in that area, she may understand it like the cited papers develop best-of-both-worlds algorithms for dueling bandit (which is of course incorrect). Also, I suggest focusing mostly on works that study adversarial settings.
- Regarding corrupted models: note that Zimmert and Seldin (2019) also cover that setting.
- The paper (Awerbuch & Kleinburg, 2008) has incorrect information. It is a COLT 2005 paper, with journal version in 2008. It says COLT 2008, which is incorrect.
- (Alatur et al., 2020a) and (Alatur et al., 2020b) are identical.
- The discussion on multiplayer adversarial bandit in the related work is very limited. Some related work are missing, e.g., "Kanade, V., Liu, Z., & Radunovic, B. (2012). Distributed non-stochastic experts. Advances in Neural Information Processing Systems, 25."

**Audience:**

No

**Audience Explanation:**

While online learning, under both bandit and full-information feedback, in multiplayer settings may be of interest to a portion of the TMLR audience, the current presentation and results are unlikely to be considered sufficiently relevant or compelling for the venue.

Although certain aspects of the setting are worthwhile, the overall problem formulation and the strength of the findings fall below the typical standard expected of technical papers at TMLR.

**Broader Impact Concerns:**

The paper studies a theoretical problem. Therefore, I believe there is no concern associated to the developments and results reported in the paper.

**Claims And Evidence:**

Yes

**Claims Explanation:**

Yes, in the sense that the claimed regret bounds are supported by mathematical proofs, which appear to be correct.

However, as discussed above, parts of the problem formulation, particularly Problem B, are not well-justified within the adversarial setting. This raises concerns about the extent to which the theoretical guarantees meaningfully support the claims in that context.

As such, the assessment is mixed.

**Requested Changes:**

See above.

---

> ### Author Response · Authors · 2026-05-15
> **response to reviewer**
>
> # Response to Reviewer DFzA
>
> We thank the reviewer for the careful read. The review caught several real errors — an incorrect EXP3 attribution, a wrong rate for Bubeck & Slivkins (2012), a K↔M swap in Theorem 2, and a duplicate .bib entry for Alatur et al. (2020) that rendered as "2020a/2020b". All are fixed in the revised PDF, along with the smaller presentational points. Below we focus on the items where we want to discuss substance.
>
> **Summary of changes in the revised PDF.** (a) §2 retitled "Single-Player Adversarial **Full-Information** Setting" with note that the paper studies full-information feedback despite the loose use of "bandit"; (b) regret definition now includes the outer expectation; (c) "multi-player"→"multiplayer" and Reg_T→R_T normalized; (d) ⟨ℓ,p⟩_t defined at first use; (e) MAB/MMAB abbreviations removed; (f) [0,1]^{K^M}→[0,1]^A; (g) redundant restatement of A condensed; (h) **Theorem 2: K↔M swap fixed** in statement and proof; (i) two-player loss table relabeled with explicit (P1,P2) axes; (j) learning rate η=√(M log K / T) now stated in Theorems 3 and 4; (k) "reward is sub-Gaussian"→"noise is sub-Gaussian" in §5; (l) EXP3 attributed to Auer et al. (1995), Õ(T)→Õ(√T) for Bubeck & Slivkins (2012), Zimmert & Seldin (2019) noted to cover the corrupted regime; (m) Kveton (2015) flagged as stochastic, Zoghi/Komiyama as stochastic vs. Saha as adversarial; (n) duplicate Alatur entry consolidated and the sentence treating them as distinct papers corrected; (o) Awerbuch & Kleinberg .bib fixed to COLT 2005, pp. 233–248; (p) Kanade, Liu & Radunovic (2012) added.
>
> **Technical novelty.** We respectfully push back on this framing. We agree running Hedge over the joint action space is a routine recipe; the contribution we want to highlight is *not* that, but: (i) the dimensional gap between the (K−1)M-dimensional product-distribution space and the full K^M simplex, which the lower bound shows an adaptive adversary can exploit even at M=K=2 (the reviewer kindly calls this "interesting"); (ii) the Bayes-rule argument that the *sequential* mHedge distribution *exactly equals* the single-player Hedge distribution over the joint space, enabling direct inheritance of single-player guarantees; (iii) for mHedge-B, the product over players no longer telescopes after sub-Gaussian noise is introduced (see the displayed inequality involving exp(2Mη√(2(t−1)log(TK^M))) on p. 9) — the good-event split and per-player noise correction yield the extra M factor. The analyses are short by design, not by triviality.
>
> **Problem B formulation.** Problem B is a modeling choice, not a technical convenience: a centralized adversary chooses ℓ_t while players observe private noisy estimates (sensors, channels with independent noise). The adversarial losses are unrestricted; sub-Gaussianity is on the *observation noise*, which is standard for heterogeneous private feedback. Removing this assumption while preserving learnability is a natural open question, which we now flag in future work.
>
> **"Bandit" terminology.** The reviewer is right that the feedback model is full-information. We've corrected this in Section 2 and throughout the technical sections. We've kept "bandit" in the title because the problem sits in the bandit literature and the term is often used loosely there, but we're happy to retitle (e.g., "...Multiplayer Adversarial Online Learning under Information Asymmetry") if preferred.
>
> **Section 4 header "no information asymmetry".** We'd like to keep this. The paper uses "information asymmetry" as an umbrella for action and reward asymmetry (introduced in §1), and §4 shows that sequential play resolves action asymmetry — so Problem A genuinely has no information asymmetry of either kind.
>
> **Theorem 1 / a_{i,j} notation.** The proof's first line defines "a_{i,j} represents player 1 selecting action i and player 2 selecting action j" — there is no belief or counterfactual, and the assertion is non-trivial (it bounds regret against *any* product distribution). We suspect the confusion came from the subsequent loss table's a_{1,·} / a_{·,1} labels; we've replaced these with explicit (P1, P2) axes labeled i=1,2 and j=1,2.
>
> **Section ordering.** We considered presenting both protocols up front but think the current order serves the argument: §3's impossibility for simultaneous play *motivates* introducing successive pulls in §4. We've added a forward-pointer in §2 mentioning both protocols.
>
> **Audience fit.** TMLR's stated criteria are technical correctness and informativeness to *some* portion of the audience, not novelty or impact. To our knowledge, multiplayer adversarial online learning under information asymmetry has not been studied in this form, and we provide both the impossibility result for simultaneous play and matching Õ(√T) upper bounds for sequential play. We hope the revised manuscript meets the bar.
>
> Thanks again for catching the genuine errors.